# Cell-Free Tumor DNA Detection-Based Liquid Biopsy of Plasma and Bile in Patients with Various Pancreatic Neoplasms

**DOI:** 10.3390/biomedicines12010220

**Published:** 2024-01-18

**Authors:** Mark Jain, David Atayan, Tagir Rakhmatullin, Tatyana Dakhtler, Pavel Popov, Pavel Kim, Mikhail Viborniy, Iuliia Gontareva, Larisa Samokhodskaya, Vyacheslav Egorov

**Affiliations:** 1Medical Research and Educational Center, Lomonosov Moscow State University, 119992 Moscow, Russia; slm@fbm.msu.ru; 2Joint Stock Company “Ilyinsky Hospital”, 143421 Moscow, Russia; d.atayan@ihospital.ru (D.A.); t.dakhtler@ihospital.ru (T.D.); p.popov@ihospital.ru (P.P.); drpoul@yandex.ru (P.K.); m.vyborniy@ihospital.ru (M.V.); y.gontareva@ihospital.ru (I.G.); v.egorov@ihospital.ru (V.E.); 3Department of Fundamental Medicine, Lomonosov Moscow State University, 119991 Moscow, Russia; tagir.rakhmatullin@internet.ru

**Keywords:** liquid biopsy, *KRAS* mutations, pancreatic ductal adenocarcinoma, pancreatic cancer, intraductal papillary mucinous neoplasm, digital droplet polymerase chain reaction, plasma, bile, prognosis, survival

## Abstract

The key challenge of cell-free tumor DNA (cftDNA) analysis in pancreatic ductal adenocarcinoma (PDAC) is overcoming its low detection rate, which is mainly explained by the overall scarcity of this biomarker in plasma. Obstructive jaundice is a frequent event in PDAC, which enables bile collection as a part of routine treatment. The aim of this study was to evaluate the performance of *KRAS*-mutated cftDNA detection-based liquid biopsy of plasma and bile in patients with pancreatic neoplasms using digital droplet PCR. The study included healthy volunteers (*n* = 38), patients with PDAC (*n* = 95, of which 20 had obstructive jaundice) and other pancreatic neoplasms (OPN) (*n* = 18). The sensitivity and specificity compared to the control group were 61% and 100% (AUC-ROC—0.805), and compared to the OPN group, they were 61% and 94% (AUC-ROC—0.794), respectively. Bile exhibited higher cftDNA levels than plasma (248.6 [6.743; 1068] vs. 3.26 [0; 19.225] copies/mL) and a two-fold higher detection rate (*p* < 0.01). Plasma cftDNA levels were associated with distant metastases, tumor size, and CA 19-9 (*p* < 0.05). The probability of survival was worse in patients with higher levels of cftDNA in plasma (hazard ratio—2.4; 95% CI: 1.3–4.6; *p* = 0.005) but not in bile (*p* > 0.05). Bile is a promising alternative to plasma in patients with obstructive jaundice, at least for the diagnostic purposes of liquid biopsy.

## 1. Introduction

Based on GLOBOCAN 2020 data, pancreatic cancer (PC) is ranked 14th in frequency and 7th in mortality amongst all oncological diseases [1]. It is estimated that by 2030, PC may be ranked significantly higher in terms of mortality, being second only to lung cancer [2]. The diagnosis of this disease in early stages is challenging due to the lack of specific symptoms and efficient screening strategies. In fact, only 5% of PC cases are diagnosed early, whereas most patients present with locally advanced or metastatic diseases [3]. Therefore, according to the latest data, the relative 5-year survival rate for all stages of PC is quite low—only 11.6%, often making it a fatal disease [4].

Pancreatic ductal adenocarcinoma (PDAC) is the most frequent neoplasm of the pancreas, accounting for approximately 90% of cases [5]. Other neoplasms of this organ include malignant tumors (such as endocrine and neuroendocrine cancer), benign tumors (such as serous cystadenoma), and premalignant tumors (such as intraductal papillary mucinous neoplasm (IPMN)), which often require continuous monitoring [6,7].

Due to both the development of sensitive analytical techniques and advances in the molecular characterization of tumor cells, a new promising approach for cancer detection and surveillance has emerged, namely, liquid biopsy—a set of methods for the analysis of tumor derivatives, such as cell-free tumor DNA (cftDNA), circulating tumor cells, tumor-educated platelets, and others, in various biological fluids [8,9,10,11].

Fortunately, PDAC is rather unique, since in up to 90% of cases, the tumor carries activating mutations in the *KRAS* gene in the same hotspots (codons 12, 13, and 61), making cftDNA detection based on the presence of these mutations quite appealing for the purposes of liquid biopsy, mainly due to the fact that they can be analyzed not only by means of next-generation sequencing technologies, but also using cheaper targeted approaches such as polymerase chain reaction (PCR) [12]. Moreover, *KRAS* mutations are considered to be early events in tumor formation [13]. Therefore, cftDNA carrying these genetic alterations may be detectable at all stages of the disease.

The key challenge of this kind of liquid biopsy in PDAC is the fact that the discovery rate of *KRAS*-mutated cftDNA in plasma is usually lower than in matching tumor tissue [14,15,16]. This might be primarily explained by overall low levels of cftDNA, which are often below 5 molecules per mL of plasma [17]. Hence, the probability of false-negative results of the analysis is quite high. In terms of cftDNA content, bile may be a promising alternative to plasma, as this biomaterial has an overall smaller volume, and it is in a state of continuous direct contact with the surface of the tumor in case of duct invasion. About 80% of PDACs are located in the head of the organ, of which approximately half are accompanied by obstructive jaundice requiring biliary drainage [18,19]. Thus, in many cases, the collection of this biomaterial may be carried out naturally as a part of routine treatment.

The aim of this study was to evaluate the diagnostic and prognostic value of *KRAS*-mutated cftDNA detection-based liquid biopsy of plasma and bile in patients with various pancreatic neoplasms.

Digital droplet PCR (ddPCR) was the method of choice for cftDNA quantification due to its resilience to PCR inhibitors, which may appear in DNA samples isolated from bile, as well as to its high analytical sensitivity, which allows it to surpass its rivals such as real-time PCR and various sequencing technologies [20,21].

## 2. Materials and Methods

### 2.1. General Information

The study was approved by the institution’s Local Ethics Committee (#12/21, 13 December 2021) and conducted according to the tenets of the Declaration of Helsinki. Patient enrollment was conducted in a private hospital (from December 2021 to January 2023). All participants provided signed informed consent forms. The study included 95 patients (of which 14 took part in our previously published pilot study [22]) with PDAC; 18 patients with various other pancreatic neoplasms (OPN) such as IPMNs (low-grade dysplasia), solid pseudopapillary neoplasm, serous cystadenoma, and others (OPN group); 38 healthy volunteers (control group) without any known oncological diseases. Among the PDAC group patients, 20 had obstructive jaundice.

Morphological verification of the diagnosis was performed using a core needle biopsy of the primary tumor or liver metastasis with histological evaluation. Endoscopic ultrasonography-guided fine-needle aspiration cytology was used in cases of localized neoplasms. The visualization and evaluation of spatial parameters of tumors were carried out using computed tomography, magnetic resonance imaging, and positron emission tomography/computed tomography with 18FDG according to medical indications. All relevant data were reevaluated by a single specialist in the respective field to eliminate the influence of interobserver variability. Serum CA 19-9 was assessed via the chemiluminescent immunometric method on the Access 2 Immunoassay System using the Access GI Monitor reagents pack (Beckman Coulter Diagnostics, Ltd., Brea, CA, USA). The demographic and clinical characteristics of study participants are summarized in Table 1, whereas more detailed data are available in Appendix A.

### 2.2. Biomaterial Collection and Processing

All study participants donated 10–12 mL of peripheral venous blood, which was collected into tubes containing EDTA. Biomaterial was collected prior to any invasive diagnostic procedures and treatment. The samples were stored at a temperature of +4 °C for no more than 4 h. Then, plasma was separated via centrifugation (3000× *g*, 10 min). The resulting plasma samples were centrifuged once more (3000× *g*, 10 min) to ensure the depletion of any cell debris. The biomaterial was stored in fresh tubes at a temperature of −80 °C.

Patients presenting PDAC and obstructive jaundice at the time of inclusion in the study underwent external biliary drainage or stenting of the common bile duct (*n* = 20). In these cases, bile samples of ~15 mL were collected and immediately frozen at a temperature of −80 °C.

After defrosting, both plasma and bile samples were thoroughly mixed by pulse-vortexing. Additionally, bile samples underwent two rounds of centrifugation at conditions described above for peripheral venous blood. Cell-free DNA (cfDNA) was isolated from 5 mL of plasma and 5 mL of bile supernatant using the QIAamp Circulating Nucleic Acid Kit (Qiagen GmbH, Hilden, Germany) with carrier RNA according to the instruction manual. However, for the bile supernatant, the lysis stage was extended for additional 30 min. In all cases, the DNA elution volume was 50 µL.

### 2.3. DNA Analysis

Droplet generation and readings were carried out using the QX200 AutoDG ddPCR System, and the amplification was performed on the CFX96 Touch instrument (Bio-Rad Laboratories, Inc., Hercules, CA, USA). All ddPCR-related manipulations were conducted according to the manufacturer’s instructions.

cftDNA detection was carried out using the ddPCR KRAS G12/G13 Screening Kit and ddPCR KRAS Q61 Screening Kit (Bio-Rad Laboratories, Inc., Hercules, CA, USA). These reagents enabled the detection of the following *KRAS* mutations: G12A, G12C, G12D, G12R, G12S, G12V, and G13D, and Q61H (183A > C), Q61H (183A > T), Q61K, Q61L, and Q61R, respectively. Screening kits for both hotspots were designed to analyze DNA on a “single tube” basis, which means that the probes for all mutant alleles were labeled using the same fluorescent dye. Therefore, during data interpretation, it was possible to define the mutated hotspot in the *KRAS* gene (G12/13 or Q61) but not the exact substitution (e.g., G12S or G12D). The following thermocycling protocol was used: incubation at 95 °C (10 min); 40 cycles of denaturation at 94 °C (30 s) and annealing/extension at 55 °C (1 min); and incubation at 98 °C (10 min). The input of DNA solution into final ddPCR reaction mixture was set to a maximum possible volume—9.9 µL. The cutoff value for the ddPCR false-positive droplets was established in a series of experiments with different loads of wild-type DNA as described previously [22] and subtracted from the cftDNA analysis results.

### 2.4. Statistical Analysis

Data were analyzed using IBM SPSS Statistics 26.0 Software (IBM Corp., Armonk, NY, USA). The results of quantitative cftDNA analysis were presented in the following two forms: as the cftDNA level (copies of mutant allele per mL of biomaterial) and as the mutant allele fraction (MAF).
MAF (%) = C_mutated DNA_/(C_wild type DNA_ + C_mutated DNA_) × 100%,
where «C» is the DNA level expressed in copies per mL of the biomaterial.

Data distribution normality was assessed using Shapiro–Wilk’s test. Due to the absence of normal distribution for all tested variables, non-parametrical statistical tests were used. Quantitative and qualitative paired data were compared using the Wilcoxon test and McNemar’s test, respectively, whereas unpaired quantitative and qualitative data were compared using the Mann–Whitney U test and Fisher’s exact test, respectively. Quantitative data are presented as medians [quartile 1; quartile 3]. Receiver operating characteristic (ROC) analysis was carried out to evaluate the performance of binary classification for certain variables. Spearman’s rank correlation coefficient (r_s_) was used to assess associations between two variables, and the size of the effect was evaluated using Chaddock’s scale. The survival analysis was conducted using Kaplan–Meier plots and the log-rank (Mantel–Cox) test. Cox proportional hazards regression analysis was used to evaluate hazard ratios (HRs) and the corresponding 95% confidence intervals (CIs). A *p*-value < 0.05 was considered statistically significant.

## 3. Results

### 3.1. Diagnostic Performance of cftDNA Analysis

Cell-free DNA analysis was successfully carried out in all samples of the 151 study participants. Detailed ddPCR analysis results for each individual sample are available in Appendix A. In the control group as well as in the most of OPN group plasma samples, none of the studied *KRAS* mutations were detected in cfDNA. There was a single case of cftDNA positivity (mutation in the *KRAS* G12/13 hotspot) in the OPN group, specifically in the plasma sample from a patient diagnosed with a pancreatic serous cystadenoma (however, the cftDNA level was extremely low—2.34 copies/mL with a MAF of 0.12%). On the contrary, the detection of cftDNA in plasma samples from the PDAC group occurred quite often—in 58 out of 95 cases. ROC curves illustrating the diagnostic potential of *KRAS* mutations detection-based plasma liquid biopsy are presented in Figure 1. At a cftDNA cutoff value of 0 copies/mL, the overall sensitivity and specificity compared to the control group were 61% and 100% (area under the ROC curve of 0.805), and compared to the OPN group, they were 61% and 94% (area under the ROC curve of 0.794), respectively.

The plasma cftDNA levels and MAFs varied significantly in the PDAC group: 5.08 [0; 17.55] copies/mL and 0.09 [0; 0.62] % with maximum values as high as 36,840 copies/mL and 50.87%, respectively. *KRAS* G12/13 was the most frequently mutated hotspot—55 out of 58 cftDNA positive cases. Mutations in the KRAS Q61 hotspot were detected only in four plasma samples (in one case, both hotspots were mutated).

Compared to the paired plasma samples, bile exhibited significantly higher cftDNA levels (3.26 [0; 19.225] vs. 248.6 [6.743; 1068] copies/mL, respectively, *p* = 0.001; Figure 2a) and MAFs (0.045 [0; 0.413] vs. 1.74 [0.2; 11.11] %, respectively, *p* = 0.002; Figure 2b). The cftDNA detection rate in bile was also marginally higher than in paired plasma samples (18/20 vs. 11/20, respectively; Figure 2c). Again, the detection of mutations in the *KRAS* Q61 hotspot was rare (only a single case in bile). It is worth mentioning that there was not a single case where a plasma sample was positive for cftDNA but the corresponding bile sample was not. No statistically significant correlations were observed for cftDNA levels and MAFs in paired plasma and bile samples (*p* > 0.05; Appendix A).

### 3.2. Relation of cftDNA to Clinical and Demographic Data

Plasma cftDNA levels were associated with the size of the tumor, which was not the case for cftDNA measured in bile (Figure 3). It appeared that in patients with smaller tumors (<2 cm), cftDNA was mostly undetectable (both the cftDNA level and MAF were 0 [0; 0] copies/mL/%) in contrast to that in patients with larger tumors (*p* < 0.05). However, there was no difference between plasma samples corresponding to tumors 2–4 cm and >4 cm in size (5.78 [0; 33.54] vs. 3.495 [0; 17.47] copies/mL—for cftDNA levels, 0.13 [0; 0.523] vs. 0.045 [0; 1.28] % for MAFs, respectively, *p* > 0.05).

The results of cftDNA analysis (regarding both levels and MAFs) in plasma correlated with serum levels of another routinely analyzed in patients with a PDAC biomarker, CA 19-9, although the effect was weak to moderate (r_s_ of 0.302, *p* = 0.003 and r_s_ of 0.228, *p* = 0.026, respectively; Appendix A). There were no statistically significant correlations for cftDNA analyzed in bile (*p* > 0.05). It is known that serum levels of CA 19-9 are not informative in a large proportion of PDAC patients due to various reasons (including the Lewis phenotype [23]). In the present cohort, 47 out of 95 PDAC patients had serum CA 19-9 levels below the cutoff value of 34 U/mL. However, the cftDNA detection rates were equal among patients with normal and elevated CA 19-9 levels, both for plasma and bile (25/47 vs. 33/48 and 9/9 vs. 9/11 cases, respectively, *p* > 0.05).

Notably, patients with distant metastases had significantly higher plasma cftDNA levels and MAFs, compared to the rest of the PDAC group (3.92 [0; 21.6] vs. 2.195 [0; 23.725] copies/mL, *p* = 0.01 and 0.1 [0; 0.65] vs. 0.025 [0; 0.413] %, *p* = 0.015, respectively; Figure 4). At a cutoff value of 91.5 copies/mL for plasma cftDNA levels, sensitivity, specificity, and positive and negative predictive values were 20.9%, 100%, 100.0 [95% CI: 66.4–100.0] % and 56.4 [95% CI: 52.6–60.2] %, whereas, at a cutoff value of 1.18% for plasma MAFs, they were 30.2%, 93.2%, 81.3 [95% CI: 57.0–93.4] % and 57.8 [95% CI: 52.5–62.8] %, respectively, highlighting the potential usefulness of plasma cftDNA analysis for the detection of distant metastases in a certain proportion of patients with PDAC. Once again, no significant associations were found in the bile samples.

Localization of the tumor in the pancreas, its contact with arteries and veins, invasion into the bile ducts, sex, and age were among the other tested variables. In neither of the cases, any statistically significant associations with cftDNA analysis results were observed (*p* > 0.05; Appendix A). It is worth noting that the results of bile analysis were not considered for evaluation based on certain tumor localization characteristics, given that this biomaterial was collected only in cases of biliary drainage due to obstructive jaundice, and hence, mostly the tumors were situated in the head of the pancreas while being in contact with blood vessels and bile ducts.

### 3.3. Prognostic Performance of cftDNA Analysis

In this study, the follow-up time after biomaterial collection was up to 2 years. Kaplan–Meier plots for PDAC group patients are presented in Figure 5. It appeared that the probability of survival was significantly lower in patients with detectable cftDNA in plasma, compared to those in whom it was undetectable (mean survival: 11.6 ± 1.1 vs. 16.1 ± 1.5 months, respectively). The prognosis (overall survival) for patients based on this qualitative parameter was characterized by an HR of 2.2 (95% CI: 1.1–4.4), *p* = 0.024. The implementation of a cutoff value for cftDNA level in plasma of 4.735 copies/mL (or 0.065% for MAF), which was established using ROC analysis, allowed us to slightly increase the prognostic performance of the cftDNA analysis (mean survival: 10.9 ± 1.2 vs. 15.9 ± 1.3 months, respectively; HR of 2.4 (95% CI: 1.3–4.6), *p* = 0.005 for patients with plasma cftDNA levels higher than either of the above-mentioned cutoff values). Differences in the probability of survival were not significant for cftDNA measured in bile (*p* > 0.05).

## 4. Discussion

In the past decade, a plethora of studies exploring the diagnostic and prognostic value of various forms of liquid biopsy in PC patients have been conducted [17,24,25,26]. Among the many advantages of this approach, it is often its potential for use in early diagnosis that is highlighted. However, it is quite challenging to truly access the value of liquid biopsy as an early diagnosis tool, given the fact that the overwhelming majority of PC patients present with advanced stages of the disease [27]. In this regard, nation-wide biobanking programs may be exceptionally useful. For example, in the recent Golestan Cohort Study by Hosen et al., it was demonstrated that urinary cftDNA carrying certain *TERT* promoter mutations is detectable up to 10 years prior to the clinical manifestations of bladder cancer [28].

An alternative way to explore the potential of liquid biopsy for early diagnosis is to access the mutation status of circulating cfDNA in patients with precancerous lesions, namely IPMNs. Even though the transition from IPMN to cancer is not a universally frequent event, patients with these neoplasms require continuous monitoring to the point of high-grade dysplasia formation (which ends in invasive cancer), when, generally, surgical removal is appropriate [29,30,31]. It is known that *KRAS* mutations are prevalent in IPMNs (approximately 50% of cases) [32,33]. Recent data obtained from a mice IPMN model suggest that *KRAS* activation promotes the formation of the neoplasm [34], although in humans, there is no link between the presence of *KRAS* mutations and progression of IPMN [33,35]. Could it be that the appearance of cftDNA carrying these genetic alterations in the bloodstream rather than the mutation status of the neoplasm itself is indicative of its cancerous transformation? In our cohort, *KRAS*-mutated cftDNA was not identified in any of the 14 patients with IPMN (low-grade dysplasia). These results are concordant with the limited data available in the literature: 0/21 patients with low-grade dysplasia (*KRAS* G12D/G12V); 0/7, 1/16, and 1/11 patients with low-grade, high-grade dysplasia, and invasive carcinoma, respectively (*KRAS* G12/13 hotspot) [36,37]. Therefore, at this stage, it might be concluded that even if *KRAS*-mutated circulating cftDNA is indicative of the cancerous transformation of IPMN, it would be true only in later stages of the malignant process than at the point of IPMN with high-grade dysplasia/invasive carcinoma at which surgical removal is generally performed.

In the OPN group, cfDNA positivity for *KRAS* mutation was detected only once, unexpectedly, in a patient with serous cystadenoma, for which alterations in this gene are not characteristic [38]. As it was previously mentioned, the cftDNA level and MAF in this case were exceptionally low, yet these results were reproducible; thus, the possibility of contamination was rejected. It is known that the *KRAS* gene is mutated in a variety of neoplasms, including tumors in lungs, the stomach, the colon, and the endometrium [39]. Therefore, in this case, the detection of *KRAS*-mutated circulating cftDNA might be a sign of a different, yet undiscovered, disease. We plan to monitor this patient and provide an update if relevant data become available.

Another advantage of cftDNA detection-based liquid biopsy is undoubtedly its prognostic potential. As was observed in this study, the negative impact of the presence of *KRAS*-mutated circulating cftDNA in the PDAC group on overall survival is concordant with the findings of other studies on the topic [40,41,42]. According to our results, this type of liquid biopsy is even more informative regarding overall survival if a certain cftDNA level/MAF cutoff value is implemented in contrast to the basic qualitative approach. The association of cftDNA with negative prognosis might be, in part, explained by its ability to reflect the ongoing metastasis (which is known to be a major cause of death in PDAC patients [43,44]), as it was demonstrated in our cohort as well as in several other studies [45,46,47]. Moreover, potential clinical applications of liquid biopsy in PDAC might go way beyond survival prediction, as this method is proving to be useful in the monitoring of response to chemotherapy and post-treatment recurrence of the disease [48,49].

It is worth mentioning that there might be an unexpected issue which hampers the prognostic performance of cftDNA analysis, not only in PDAC but in other cancers as well, namely, the unit selection dilemma. In the present study, we chose to report our data in two forms (cftDNA level in copies/mL and MAF in %) specifically to address it. On the one hand, the relative approach for unit selection (MAF in %) has a plethora of advantages. Mainly, it allows us to easily make comparisons of the results between studies as well as to diminish the influence of the sample processing (e.g., efficacy of cfDNA isolation) and individual variations in cfDNA degradation speed in the bloodstream. On the other hand, it is known that tumor growth is accompanied by hypoxia and even death of the surrounding cells (needless to say that the former is actively occurring during the chemotherapy), which causes an increase in the total cfDNA level [50,51], which in turn influences MAF and masks its alterations. Looking at our data, it might be noted that the *p*-values across the most clinically relevant comparisons were slightly more significant in the case of the cftDNA level rather than MAF. This effect could be even more prominent if the study design included any form of prolonged monitoring, especially in a chemotherapy cycle. Possibly, the selection of a specific stable internal or external marker to normalize the cftDNA level could resolve this issue and improve the performance of this type of liquid biopsy in cancer.

Also, this study aimed to address another weak point of cftDNA detection-based liquid biopsy—the limited discovery rate of the mutant allele, which occurs mainly due to low levels of circulating cftDNA in general [17]. According to our results, bile was superior to plasma as a source of cftDNA in PDAC, both in terms of its levels and detection rates. Moreover, in our cohort, the latter in bile reached 90%, which is the expected frequency of *KRAS* mutations in patients with PDAC (based on tumor tissue sequencing studies [12]). These results are supported by findings of other authors who demonstrated that the cftDNA levels in the bile of patients with various biliary tract cancers and PDAC are also significantly higher than in plasma (by 21–64%) [52,53]. The abundance of tumor-derived genetic material in bile may provide opportunities for cfDNA analysis using cheaper and less sensitive techniques such as Sanger sequencing and real-time PCR, which could be of interest for the purpose of risk stratification based on tumor genotyping [54]. However, according to our data, it appears that bile cftDNA levels and MAFs are primarily dependent on the localization of the tumor and thus are not reflective of the studied prognostically relevant parameters. Another drawback of bile liquid biopsy is obviously the inconvenience of the collection of this biomaterial in patients who do not require biliary drainage, as it would probably imply either percutaneous aspiration or endoscopy-guided aspiration. Nevertheless, these procedures in PDAC patients still appear to be safer than any form of direct tumor biopsy, thus more appealing, at least, for cftDNA detection/genotyping purposes [55,56,57].

This study had certain limitations. Firstly, the control group consisted of healthy volunteers who were younger compared to other groups (the purpose of this group was mainly to ensure that there were no false-positive droplets in samples with wild-type DNA), whereas the OPN group was rather small. In clinical application, the analysis would primarily have been carried out in differential diagnosis of patients with certain pancreatic diseases. Secondly, we were able to obtain bile only from a limited number of patients, which could have influenced the significance of statistical analyses for data corresponding to this biomaterial. Thirdly, the mutational status of cfDNA was not compared to that of the paired tumor genomic DNA, which may be useful to verify the absence of false mutation calls. However, the exceptionally high prevalence of *KRAS* mutations in PDAC patients, in general, as well as the robust determination of false-positive signal cutoff values (resulting in 100% specificity) at least, to some extent, compensate for this. Lastly, due to limitations in the availability of clinical data for some patients, we were unable to determine the progression-free survival in the cohort and thus present only the overall survival.

## 5. Conclusions

Tumor cell-free detection-based liquid biopsy is a promising minimally invasive tool for diagnostic and prognostic applications in PDAC. The fact that the vast majority of tumors harbor *KRAS* mutations in the same hotspots allows us to utilize cheaper targeted genetic analysis techniques such as ddPCR, which provide broad potential for clinical implementation. The demonstrated superiority of bile, in the present study, compared to plasma in terms of both cftDNA detection rates and levels may promote further studies which could explore the usefulness of this biomaterial for the purposes of liquid biopsy with other tumor-derived targets. Nevertheless, there are still certain challenges to overcome for this approach to be widely used in patients with pancreatic neoplasms, namely, the standardization of liquid biopsy-related procedures, cost and time reduction of the analysis, as well as an assessment of potential clinical benefits upon its introduction into daily practice based on the results of interventional studies.

## Figures and Tables

**Figure 1 biomedicines-12-00220-f001:**
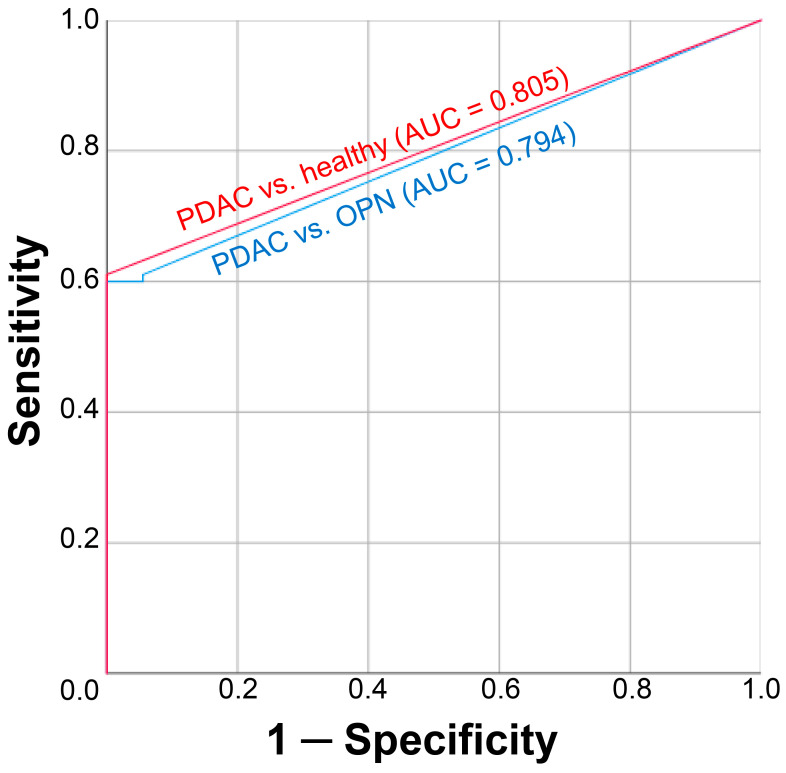
ROC curves for plasma cell-free tumor DNA analysis. Red curve corresponds to the discriminating potential for patients with PDAC and healthy volunteers, whereas blue curve corresponds to patients with PDAC and patients with OPN. PDAC, pancreatic ductal adenocarcinoma; OPN, other pancreatic neoplasms (the list of neoplasms in this group is presented in Table 1). Cell-free tumor DNA analysis was based on the detection of various mutations in *KRAS* G12/G13 and Q61 hotspots using ddPCR.

**Figure 2 biomedicines-12-00220-f002:**
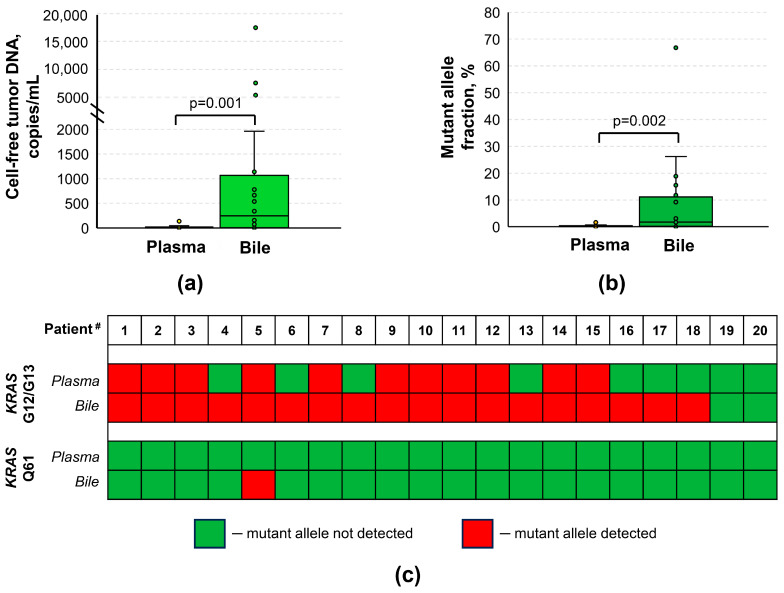
Comparison of cell-free tumor DNA analysis results in paired samples of plasma and bile of patients with pancreatic ductal adenocarcinoma. (**a**) Box plot for data presented as copies per 1 mL of biomaterial. (**b**) Box plot for data presented as mutant allele fraction. (**c**) Detection statuses for mutations in both hotspots in individual samples. #, numbers in the figure correspond to patient numbers in Appendix A. Cell-free tumor DNA analysis was based on the detection of various mutations in *KRAS* G12/G13 and Q61 hotspots by means of ddPCR.

**Figure 3 biomedicines-12-00220-f003:**
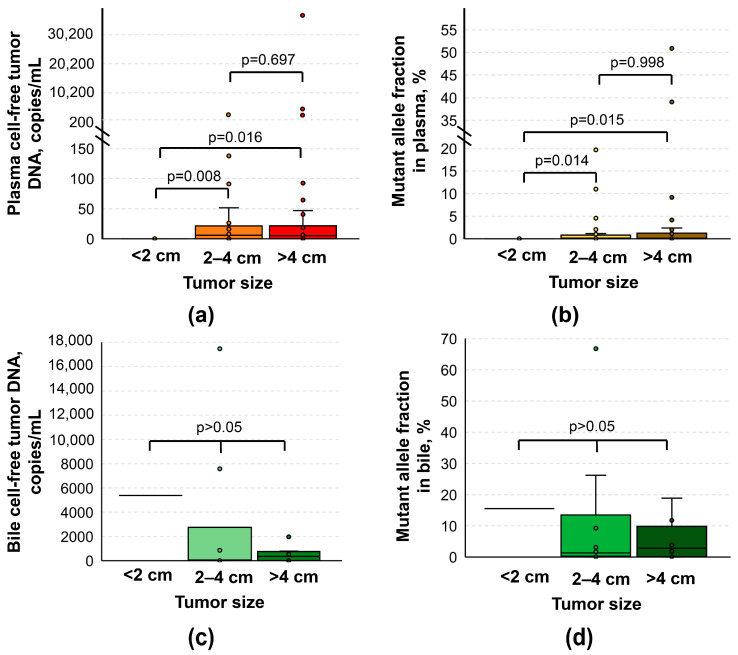
Comparison of cell-free tumor DNA analysis results based on tumor size. (**a**) Box plot for data presented as copies per 1 mL of plasma. (**b**) Box plot for data presented as mutant allele fraction in plasma. (**c**) Box plot for data presented as copies per 1 mL of bile. (**d**) Box plot for data presented as mutant allele fraction in bile. Tumor sizes of <2 cm were rare events (*n* = 7 for plasma samples; *n* = 1 for bile samples). Cell-free tumor DNA analysis was based on the detection of various mutations in KRAS G12/G13 and Q61 hotspots using ddPCR.

**Figure 4 biomedicines-12-00220-f004:**
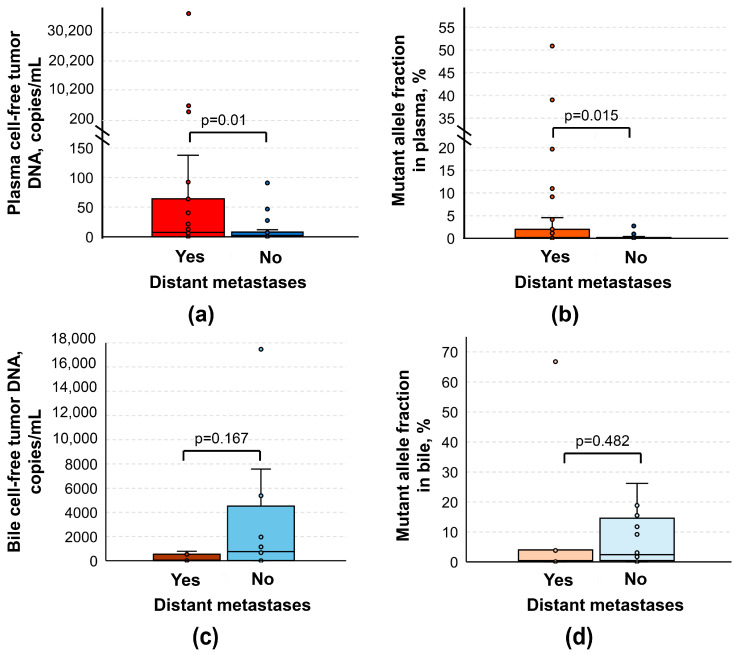
Comparison of cell-free tumor DNA analysis results in patients with and without distant metastases. (**a**) Box plot for data presented as copies per 1 mL of plasma. (**b**) Box plot for data presented as mutant allele fraction in plasma. (**c**) Box plot for data presented as copies per 1 mL of bile. (**d**) Box plot for data presented as mutant allele fraction in bile. Cell-free tumor DNA analysis was based on the detection of various mutations in *KRAS* G12/G13 and Q61 hotspots using ddPCR.

**Figure 5 biomedicines-12-00220-f005:**
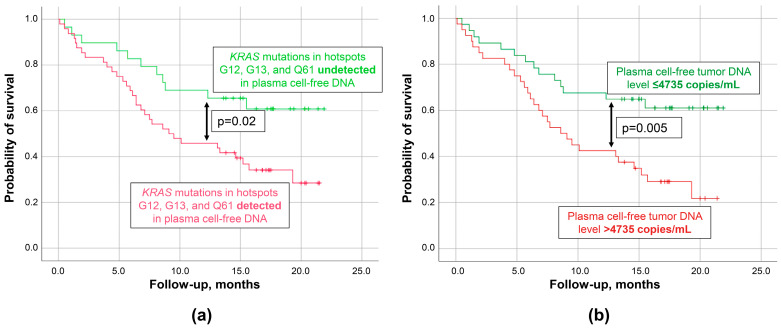
Caplan–Meier plots for plasma cell-free tumor DNA analysis in patients with pancreatic ductal adenocarcinoma. (**a**) Based on the event of detection. (**b**) Based on the surpassing of a cutoff value. Censored observations were marked with a cross. Cutoff value of 4735 copies/mL for plasma cell-free tumor DNA level was established using ROC analysis. Cutoff value for mutant allele fraction was 0.065%, which yielded identical survival curves to that of the 4735 copies/mL cutoff value. Cell-free tumor DNA analysis was based on the detection of various mutations in *KRAS* G12/G13 and Q61 hotspots using ddPCR.

**Table 1 biomedicines-12-00220-t001:** The demographic and clinical characteristics of study participants.

Parameters	PDAC Group	OPN Group	Control Group
	(*n* = 95)	(*n* = 18)	(*n* = 38)
Age, years ^1^	65 (41–88)	58 (43–66)	20 (19–22)
Sex, *n* (%):			
male	46/95	5/18	17/38
female	49/95	13/18	21/38
OPN types:			
mdIPMN, *n*	N/A	6/18	N/A
multIPMN, *n*	N/A	1/18	N/A
bdIPMN, *n*	N/A	7/18	N/A
adenoma, *n*	N/A	1/18	N/A
serous cystadenoma, *n*	N/A	1/18	N/A
SPPN, *n*	N/A	1/18	N/A
NET, *n*	N/A	1/18	N/A
Tumor localization:			
head, *n*	44/87 ^2^	N/A	N/A
body, *n*	6/87 ^2^	N/A	N/A
tail, *n*	11/87 ^2^	N/A	N/A
head + body, *n*	10/87 ^2^	N/A	N/A
body + tail, *n*	15/87 ^2^	N/A	N/A
head + body + tail, *n*	1/87 ^2^	N/A	N/A
Tumor size:			
>4 cm, *n*	39/87 ^2^	N/A	N/A
2–4 cm, *n*	41/87 ^2^	N/A	N/A
<2 cm, *n*	7/87 ^2^	N/A	N/A
Contact with			
arteries/veins, *n*	78/87 ^2^	N/A	N/A
Bile ducts invasion, *n*	39/87 ^2^	N/A	N/A
Distant metastases, *n*	43/87 ^2^	N/A	N/A
Serum CA 19-9, U/mL ^3^	36.5 [0.9; 883.3]	3.9 [0.8; 7.9]	N/A

PDAC, pancreatic ductal adenocarcinoma; OPN, other pancreatic neoplasms; N/A, not available or not applicable; mdIPMN, main duct intraductal papillary mucinous neoplasm; multIPMN, multifocal intraductal papillary mucinous neoplasm; bdIPMN, branch duct intraductal papillary mucinous neoplasm; SPPN, solid pseudopapillary neoplasm; NET, neuroendocrine tumor; ^1^ data presented as mean (range); ^2^ data were not available for 8/95 patients; ^3^ data presented as median [quartile 1; quartile 3].

## Data Availability

The data presented in this study are available in Appendix A.

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
