# Peer review of "Cell-Free Tumor DNA Detection-Based Liquid Biopsy of Plasma and Bile in Patients with Various Pancreatic Neoplasms"

_biomedicines, 2024, doi:10.3390/biomedicines12010220_

Round 1

Reviewer 1 Report

Comments and Suggestions for Authors

In the manuscript, the authors attempted to report the cftDNA detection-based liquid biopsy of plasma and bile in patients with pancreatic neoplasms, emphasizing the diagnostic and prognostic potential of this technique in PDAC. The authors have demonstrated that the higher detection rate and cftDNA levels in bile compared to plasma, which is particularly relevant for patients with obstructive jaundice. Overall, The authors’ claims are well supported by their data and this research has certain value on clinical applications. However, there are some concerns on the manuscript.

- My only concern is about the conclusions drawn on the future predictive value of ctDNA levels. It's evident that the highest ctDNA levels are found in patients at high risk, but the data doesn't clearly show if ctDNA levels offer any new or extra predictive information. More clear explanations or further studies might be needed to confirm these findings.

Possible Typos:

Page 4, line 130: “Screening Kit и”

Author Response

We are grateful for the time and effort dedicated to providing valuable comments on our manuscript. We have done our best to revise the manuscript and include all provided suggestions. The changes are highlighted in the revised manuscript using MS Word tools.

Here is a point-by-point response to received comments.

Comment 1: “My only concern is about the conclusions drawn on the future predictive value of ctDNA levels. It's evident that the highest ctDNA levels are found in patients at high risk, but the data doesn't clearly show if ctDNA levels offer any new or extra predictive information. More clear explanations or further studies might be needed to confirm these findings”.

Response: We are grateful for this comment. Our team agrees that “Tumor cell-free detection-based liquid biopsy is an exceptional minimally invasive tool for diagnostic and prognostic applications in PDAC” might be a premature conclusion. Therefore, “exceptional” was substituted for “promising”. Moreover, the last sentence of the conclusions was expanded: “Nevertheless, there are still certain challenges to overcome for this approach to be widely used in patients with pancreatic neoplasms, namely standardization of liquid biopsy related procedures, cost and time reduction of the analysis, as well as assessment of potential clinical benefits upon its introduction into daily practice based on results of interventional studies”. Additionally, positive and negative predictive values were calculated for cftDNA analysis regarding the presence of distant metastases (lines 232-237).

Comment 2:Possible Typos: Page 4, line 130: “Screening Kit и””

Response: Thank you for pointing this out. This typo as well as several others were corrected.

Reviewer 2 Report

Comments and Suggestions for Authors

This is a very interesting work addressing the potential use of cell-free tumor DNA (cftDNA) in both plasma and bile as a diagnostic tool in patients with obstructive jaundice. While this is not a novel investigation, the author’s work brings suffice results to support the scientific value of their study.

The utility of cftDNA as a screening tool in the diagnostic and surveillance of pancreatic cancer patients may be limited due to its low sensitivity of 61% (as per author’s results). It will be very helpful if the authors will show the positive and negative predictive values when they investigated the comparison of cftDNA analysis in patients with and without metastases as in figure 4). A high positive predictive value, despite low sensitivity, it may help to diagnose progression (including de novo metastasis diagnosis) in patients with detectable cftDNA in bile and plasma. Will be cftDNA from bile more reliable than cftDNA from plasma?

Since pancreatic tumors are known to shed lower cftDNA, we may expect most likely false negative result in plasma, in spite of presence of tumor targetable mutation. What is the perspective of the authors with regard to possible false negative results in bile (higher/lower)? Is the tumor cell shedding different in bile vs plasma, both in terms of quantity and quality of the genetic material?

The authors show in figure 4 that “patients with distant metastases had significantly higher plasma cftDNA levels and MAFs, compared to the rest of the PDAC group” (lines 228-229). Despite not reaching significance (due to the low number of samples), there is clearly opposite trend in bile vs plasma in the figure 4. While this may be unexpected, the authors must hypothesize why they got this trend in their investigation.

The authors must re-phrase their statement in Discussion section (lines 337-340).  Driescher et al’s work (reference 52 in the manuscript) demonstrated not only “that the cftDNA levels in bile of patients with gallbladder and biliary tract cancers is also significantly higher than in plasma” but also they showed the same trend for pancreatic cancer. This is an important information to first recognize the work of previous groups and second to discuss how both works can complementarily improve the investigation of cftDNA in bile for pancreatic cancer diagnosis.

As a final minor comment, there are a lot of typos in the manuscript. Please review them accordingly.

Author Response

We are grateful for the time and effort dedicated to providing valuable comments on our manuscript. We have done our best to revise the manuscript and include all provided suggestions. The changes are highlighted in the revised manuscript using MS Word tools.

Here is a point-by-point response to received comments.

Comment 1: “The utility of cftDNA as a screening tool in the diagnostic and surveillance of pancreatic cancer patients may be limited due to its low sensitivity of 61% (as per author’s results). It will be very helpful if the authors will show the positive and negative predictive values when they investigated the comparison of cftDNA analysis in patients with and without metastases as in figure 4). A high positive predictive value, despite low sensitivity, it may help to diagnose progression (including de novo metastasis diagnosis) in patients with detectable cftDNA in bile and plasma. Will be cftDNA from bile more reliable than cftDNA from plasma?”.

Response: Thank you for this suggestion. The following additions were made: “At a cutoff value of 91.5 copies/mL for plasma cftDNA levels sensitivity, specificity, positive and negative predictive values were 20.9%, 100%, 100.0 [95% CI: 66.4–100.0] % and 56.4 [95% CI: 52.6–60.2] %, whereas at a cutoff value of 1.18% for plasma MAFs – 30.2%, 93.2%, 81.3 [95% CI: 57.0–93.4] % and 57.8 [95% CI: 52.5–62.8] %, respectively, highlighting potential usefulness of plasma cftDNA analysis for the detection of distant metastases in a certain proportion of patients with PDAC. Once again, no significant associations were found in bile samples”.

Comment 2: “Since pancreatic tumors are known to shed lower cftDNA, we may expect most likely false negative result in plasma, in spite of presence of tumor targetable mutation. What is the perspective of the authors with regard to possible false negative results in bile (higher/lower)? Is the tumor cell shedding different in bile vs plasma, both in terms of quantity and quality of the genetic material?

Response: Thank you for this comment. The following sentence was added to the discussion section to highlight the fact that our results suggest that bile, in contrast to plasma, possibly allows to avoid false-negative results (at least for KRAS-mutated cell-free DNA analysis):

According to our results, bile was superior to plasma as a source of cftDNA in PDAC both in terms of its levels and detection rates. Moreover, in our cohort the latter in bile reached 90%, which is the expected frequency of KRAS mutations in patients with PDAC (based on tumor tissue sequencing [12]).

It is difficult for us to estimate the differences in the cell shedding in bile and plasma as our study aimed only to investigate the differences in cell-free tumor DNA. We did not analyze such targets as circulating tumor cells. It is known that cell-free DNA originates not only from dead cells but there is some active secretion. Therefore, we are not sure that we can compare cell shedding solely based on our results in the present study.  Also, we cannot legitimately compare the quality of cell-free DNA in bile and plasma. In case of bile we froze intact biomaterial, then upon defrosting we centrifuged it in order to collect supernatant prior to the isolation of DNA. Thus, in contrast to plasma, bile DNA samples resembled not only the cell-free component of the biomaterial. Moreover, we used the cell-free DNA isolation kit “off-label” with minor adjustments of the protocol, as there are no kits for bile cell-free DNA isolation available. These factors can severely impact the comparison of DNA quality (fragmentation and impurities). Nevertheless, our team agrees that a technical study dedicated to the comparison of various DNA isolation methods (in terms of both quality and quantity of the DNA) from bile is needed (especially now when more and more data appears regarding diagnostic and prognostic value of bile cell-free DNA analysis). We plan to address this issue in our future studies. The last sentence of the conclusions section highlights challenges that are yet to overcome to make this approach viable for clinical implementation:

“Nevertheless, there are still certain challenges to overcome for this approach to be widely used in patients with pancreatic neoplasms, namely standardization of liquid biopsy related procedures, cost and time reduction of the analysis, as well as assessment of potential clinical benefits upon its introduction into daily practice based on results of interventional studies”.

Comment 3:The authors show in figure 4 that “patients with distant metastases had significantly higher plasma cftDNA levels and MAFs, compared to the rest of the PDAC group” (lines 228-229). Despite not reaching significance (due to the low number of samples), there is clearly opposite trend in bile vs plasma in the figure 4. While this may be unexpected, the authors must hypothesize why they got this trend in their investigation.”

Response: Thank you for this comment. Figures 3 and 4 were modified. Now the exact p-values are displayed for the insignificant comparisons (instead of “p>0.05”, which in our case appeared to be misleading). For the comparisons of bile regarding the presence of distant metastases the p-values were in fact 0.167 and 0.482. Therefore, unfortunately, there was no trend for bile. The following sentences in the discussion section provide our point of view on the prognostic relevancy of bile cftDNA analysis: “The abundance of tumor-derived genetic material in bile may provide opportunities for the cfDNA analysis using cheaper and less sensitive techniques such as Sanger sequencing and real-time PCR, which could be of interest for the purposes of risk stratification based on tumor genotyping [54]. However, according to our data, it appears that bile cftDNA levels and MAFs are primarily dependent on the localization of the tumor, and thus are not reflective of the studied prognostically relevant parameters.”

Comment 4:The authors must re-phrase their statement in Discussion section (lines 337-340).  Driescher et al’s work (reference 52 in the manuscript) demonstrated not only “that the cftDNA levels in bile of patients with gallbladder and biliary tract cancers is also significantly higher than in plasma” but also they showed the same trend for pancreatic cancer. This is an important information to first recognize the work of previous groups and second to discuss how both works can complementarily improve the investigation of cftDNA in bile for pancreatic cancer diagnosis.

Response: Thank you for pointing out this mistake. Suggested corrections were made.

Comment 5: “As a final minor comment, there are a lot of typos in the manuscript. Please review them accordingly.”

Response: Thank you for pointing this out. Typos all over the text were reviewed. Corrections are highlighted using MS WORD tools.

Round 2

Reviewer 2 Report

Comments and Suggestions for Authors

I would like to congratulate the authors for their pertinent responses and comments/adding. I believe that the manuscript in the present form is suitable for publication. Thank you.